# CCL-11 or Eotaxin-1: An Immune Marker for Ageing and Accelerated Ageing in Neuro-Psychiatric Disorders

**DOI:** 10.3390/ph13090230

**Published:** 2020-09-02

**Authors:** Mariya Ivanovska, Zakee Abdi, Marianna Murdjeva, Danielle Macedo, Annabel Maes, Michael Maes

**Affiliations:** 1Department of Microbiology and Immunology, Faculty of Pharmacy, Research Institute, Medical University-Plovdiv, 4002 Plovdiv, Bulgaria; mariana.murdzheva@mu-plovdiv.bg; 2Laboratory of Clinical Immunology, University Hospital “St. George”-Plovdiv, 4002 Plovdiv, Bulgaria; 3Division of Immunological Assessment of Post-Traumatic Stress Disorder, TCEMED, 4002 Plovdiv, Bulgaria; 4Medical Faculty, Medical University-Plovdiv, 4002 Plovdiv, sBulgaria; zabdi08@googlemail.com; 5Neuropsychopharmacology Laboratory, Drug Research, and Development Center, Faculty of Medicine, Universidade Federal do Ceará, Fortaleza 60455-760, Brazil; daniellesilmacedo@gmail.com; 6National Institute for Translational Medicine (INCT-TM, CNPq), Ribeirão Preto 14000-000, Brazil; 7Janssen Pharmaceutica NV, 2340 Beerse, Belgium; AMaes19@its.jnj.com; 8Department of Psychiatry, Chulalongkorn University, Bangkok 10140, Thailand; dr.michaelmaes@hotmail.com; 9TCEMED, Medical University-Plovdiv, 4002 Plovdiv, Bulgaria; 10IMPACT Strategic Research Center, Deakin University, Geelong 3220, Australia

**Keywords:** brain, behaviour, cytokines, CCL-11, eotaxin, Alzheimer’s disease, aging, stroke, schizophrenia, biomarkers, prevention

## Abstract

Background: CCL-11 (eotaxin) is a chemokine with an important role in allergic conditions. Recent evidence indicates that CCL-11 plays a role in brain disorders as well. This paper reviews the associations between CCL-11 and aging, neurodegenerative, neuroinflammatory and neuropsychiatric disorders. Methods: Electronic databases were searched for original articles examining CCL-11 in neuropsychiatric disorders. Results: CCL-11 is rapidly transported from the blood to the brain through the blood-brain barrier. Age-related increases in CCL-11 are associated with cognitive impairments in executive functions and episodic and semantic memory, and therefore, this chemokine has been described as an “Endogenous Cognition Deteriorating Chemokine” (ECDC) or “Accelerated Brain-Aging Chemokine” (ABAC). In schizophrenia, increased CCL-11 is not only associated with impairments in cognitive functions, but also with key symptoms including formal thought disorders. Some patients with mood disorders and premenstrual syndrome show increased plasma CCL-11 levels. In diseases of old age, CCL-11 is associated with lowered neurogenesis and neurodegenerative processes, and as a consequence, increased CCL-11 increases risk towards Alzheimer’s disease. Polymorphisms in the CCL-11 gene are associated with stroke. Increased CCL-11 also plays a role in neuroinflammatory disease including multiple sclerosis. In animal models, neutralization of CCL-11 may protect against nigrostriatal neurodegeneration. Increased production of CCL-11 may be attenuated by glucocorticoids, minocycline, resveratrol and anti-CCL11 antibodies. Conclusions: Increased CCL-11 production during inflammatory conditions may play a role in human disease including age-related cognitive decline, schizophrenia, mood disorders and neurodegenerative disorders. Increased CCL-11 production is a new drug target in the treatment and prevention of those disorders.

## 1. Introduction

Chemokines consist of a family of small cytokines (7–12 kDa), which prompt directed chemotaxis in nearby responsive cells. Chemokines play a pivotal role in immune functions and inflammatory responses and facilitate leukocyte migration and trafficking [1,2,3]. The discovery of chemokines goes back to 1977 [4,5], but their role in altering the neuroimmune and neurobiological processes did not gain notice until the mid-1990s [6]. Recent studies show a direct role of chemokines in neuroendocrine function, neurotransmission and neurodegeneration within the CNS (Central Nervous System) [7]. Chemokines comprise four families characterized according to the relative position of their cysteine residues and their functions, with CCL (C–C motif chemokine) and CXCL (C-X-C motif Ligand) being the largest [7]. They act by binding to seven transmembrane G protein-coupled receptors, which in turn activate signaling cascades and initiate shape rearrangement and cell movement [7].

One of those chemokines, namely CCL-11 or eosinophil chemotactic protein (eotaxin), is involved in the selective recruitment of eosinophils into inflammatory sites during allergic reactions, and this chemokine is extensively examined in asthma, allergic rhinitis and other eosinophil-related conditions [8].

CCL-11 production is induced by T helper (Th)-2 cytokines, like IL-13 (Interleukin-13), IL-10 (Interleukin-10) and IL-4 (Interleukin-4). It is a product of eosinophils, B-cells, fibroblasts, endothelial cells, macrophages, chondrocytes and other cells [9,10] (Table 1 and Table 2).

CCL-11 is transported from the blood to the brain through the Blood-Brain Barrier (BBB) and also synthesized by microglia [11]. Furthermore, there is some evidence that CCL-11 is associated with aging and reduced neurogenesis [12]. Increased levels of CCL-11 have been detected in numerous neuro-inflammatory disorders such as multiple sclerosis [13], as well as neurodegenerative and neuroprogressive disorders including Alzheimer’s disease [8] and psychiatric illnesses including major depression, bipolar disorder and schizophrenia [8,11,13,14,15]. Moreover, increased CCL-11 levels are also associated with neurocognitive deficits in aging, neurodegenerative disorders and major psychiatric disorders such as schizophrenia [11]. This is important, because the association between CCL-11 and hippocampal damage in aging may be important to understand the pathophysiology of Alzheimer’s disease and old-age depression [12,16]. This paper aims to review the associations between CCL-11 and psychiatric disorders and its possible role as an immune biomarker in those disorders.

## 2. Results and Discussion

### 2.1. CCL-11 and CCR3 in Allergic Inflammation

Chemokine Receptors (CCRs) can bind to different ligands (CCLs), and chemokines can interact with more than one receptor [17]. The MCP (Monocyte Chemoattractant Protein) family of chemokines binds most often to CCR2, but MCP-2, MCP-3 and MCP-4 can also interact with CCR1 and CCR3 [17]. CCL-11 shows very high homology with the MCP family [18] and CCL-11 signals via the chemokine receptor CCR3 [19]. This receptor is expressed on eosinophils, basophils and Th-2-type lymphocytes, making it an attractive target for allergic disease therapies [19,20]. CCL-11, CCL-24 (eotaxin-2) and CCL-26 (eotaxin-3) all bind to CCR3 [21]. There is some evidence that high concentrations of CCL-11 are sufficient to activate CCR2 in chemotaxis assays and that substimulatory concentrations of CCL-11 can antagonize MCP-1 activity at CCR2, indicating that CCL-11 behaves as a partial agonist at CCR2 [22]. This is in contrast with Ogilvie et al. (2001) who described CCL-11 as a natural antagonist of CCR2 and an agonist of CCR5 [23]. CCL-11 shows a low affinity for binding with CXCR3 (C-X-C chemokine Receptor 3) expressed on Th-1 cells, but it is postulated that this binding can play a role in impaired Th-1 response in pathological conditions [24]. CCL-11 production is stimulated by IL-4, IL-13, IL-10, IL-1β and TNF-α in epithelial cells of the lung and the gastrointestinal tract or fibroblasts [25,26]. In 1994, CCL-11 was identified as a highly specific eosinophil chemokine that can be produced by lymphocytes, macrophages, bronchial smooth muscle cells, endothelial cells and eosinophils and that this chemokine is responsible for the regulation of chemotaxis through binding to the CCR3 [27].

Allergic diseases can be caused by complex interactions between Th-2 cells, mast cells, basophils and eosinophils, which all express CCR3 [28,29]. Romagnani (2002) showed that Th-2 cytokines contribute to the pathogenesis of allergic inflammation, as well as to the manifestation of allergy and asthma and that this proceeds at least in part through the expression of CCR3, which interacts with CCL-11, allowing the recruitment of basophils, eosinophils and mast cells [30]. CCL-11 plays a role in the pathogenesis of allergic airway diseases, inflammatory bowel disorder disease and gastro-intestinal allergic hypersensitivity [30]. Garcia et al. (2005) confirmed the role of CCR3 and CCL-11 (as well as CCR4, CCR8) in allergic inflammation using in vitro/in vivo experimental studies and clinical studies in patients with asthma [31]. The binding of CCL-11 (but also CCL24 and CCL26) to CCR3 is involved in the development of asthma symptoms [21].

Due to the significant role of CCR3 in allergic diseases, research has focused on treatments with chemokine receptor antagonists [32]. For example, inhibition of CCR3 to selectively inhibit eosinophil recruitment into tissue sites can have beneficial effects and be used as an effective therapy for allergic diseases [28].

### 2.2. CCL-11, the Blood-Brain Barrier and the CNS

CCL-11 is transferred from the blood to brain tissues with a slow phase of influx prior to the rapid phase [33]. The striatum shows an early rapid uptake phase, in contrast to other regions, which present with a delayed uptake phase [33]. CCL-11 may have biphasic effects with neuroprotective and neurotoxic effects, which are detected at physiological and pathological levels of this chemokine, respectively [33]. The same authors also concluded that CCL-11 does not cause a disturbance in the BBB [33]. Nevertheless, CCL-11 may downregulate, in a concentration-dependent manner, the tight junction proteins occludin, zona occludens-1 and claudin-1 in human coronary artery endothelial cells [34], suggesting that CCL-11 may also affect the BBB. In a study that examined patients with schizophrenia, significant associations between increased CCL-11 plasma concentrations and IgA levels directed to claudin-5 (an indicant of BBB breakdown) were found, suggesting that CCL-11 or associated mechanisms may affect the BBB [35].

Previous research [36] showed that CCL-11 is released by activated astrocytes while the CCR3 receptor is expressed by microglia, causing microglial production of reactive oxygen species (ROS) by upregulating NOX1, leading to excitotoxic neuronal death while inhibition of NOX1 can reverse these effects. The same authors showed that the release of CCL-11 by activated astrocytes causes oxidative stress due to microglial NOX1 activation, thereby inducing increased neurotoxicity (an event linked to the pathogenesis of various neurological disorders) [36]. Zhu et al. (2017) concluded that CCR3 is expressed by hippocampal neurons and that treatment of primary hippocampal neuronal cultures with CCL-11 (in vitro) causes activation of cyclin-dependent kinase 5 (Cdk5) and glycogen synthase kinase-3β (GSK3) [37] and that these effects could be blocked using CCR3 specific antagonists. CCR3 and CCR5 are present on microglia of both control and Alzheimer’s disease (AD) brains [38].

### 2.3. CCL-11: An Endogenous Cognitive Deteriorating Chemokine 

Villeda et al. (2011) established that, in animal models, age-associated rises in CCL-11 are associated with deficits in cognitive functions due to decreased neurogenesis and diminished hippocampal-related learning and memory. Young mice administered CCL-11 developed decreased adult neurogenesis in addition to diminished memory and learning, hence identifying CCL-11 as a chemokine that decreases hippocampal functions with increasing age [12]. However, another study could not find a direct effect of CCL-11 on neuronal cells, but established that CCL-11 promotes microglial migration and activation with subsequent production of ROS, which leads to glutamate-induced neuronal cell death [36]. Baruch et al. (2013) showed that a local (choroid plexus epithelium) shift toward Th-2 (T-helper 2) activation initiates IL-4 and subsequently CCL-11 production in association with cognitive deficits [16]. Thus, based on these findings and those of Villeda and Baruch, it may be concluded that age-related increases in CCL-11 may have detrimental effects on central neuronal functions [39]. The latter authors also confirmed that with age, CCL-11 levels rise in both plasma and Cerebral Spinal Fluid (CSF) and also in different neurodegenerative diseases [40].

Peripheral CCL-11 levels increase with age, and people with cognitive impairments tend to present with higher plasma CCL-11 levels than those without [39]. This suggests that CCL-11 could be a means of predicting cognitive impairments in older individuals [41]. In normal healthy volunteers, CCL-11 is significantly associated with age and the results of different neurocognitive probes as assessed with the neuropsychological tests of the Consortium to Establish a Registry for Alzheimer’s Disease (CERAD) [11]. More specifically, higher serum levels of CCL-11 are significantly correlated with lower scores on assessments of semantic and episodic memory, including the Verbal Fluency Test, Word List Memory, and Word List True Recall [11]. Moreover, CCL-11 was also associated with lowered scores on the Mini-Mental State Examination (MMSE) and diverse executive tests as measured with the Cambridge Neuropsychological Test Automated Battery (CANTAB), including Spatial Working Memory, which probes the task strategy employed by the central executive and executive working memory ability, and One-Touch Stockings of Cambridge (OTS), which probes spatial planning [11]. Moreover, age and CCL-11 have similar effects on all those neuro-cognitive tests, while CCL-11 is a partial mediator of the effects of age on these tests [11]. Furthermore, a “super-variable” comprising both age and CCL-11 exerted much stronger effects on these different tests. For example, this super-variable explained 75% of the variance in executive functions and 44.3% of the variance in an index of semantic memory. Therefore, these authors concluded that CCL-11 is an Endogenous Cognition Deteriorating Chemokine (ECDC) or “Accelerated Brain-Aging Chemokine” (ABAC) [11].

### 2.4. CCL-11 in Schizophrenia

Schizophrenia (SCZ) is a chronic psychiatric disorder characterized by neuroprogression, and its aetiology is multifactorial, with genetic and environmental components [42,43]. There is evidence that acute psychotic episodes, chronic schizophrenia and first-episode psychosis are associated with activated macrophage M1, Th-1, Th-2, Th-17 and T regulatory (Treg) responses [42,43,44,45].

CCL-11, as well as other cytokines/chemokines (including CCL-2, CCL-17, CCL-22) are significantly higher in schizophrenic patients as compared with controls [46]. Increased CCL-11 levels show a negative correlation with telomere length and grey matter volume [47]. Combining CCL-11 with four other biomarkers (namely sTNF-R1, sTNF-R2, IL-10 and IL-4) allows predicting the diagnosis of schizophrenia with a sensitivity of 70.0% and a specificity of 89.4% [48]. Frydecka et al. (2018) observed that schizophrenia is accompanied by simultaneous increases in CCL-11 and CCL-2, while increases in both chemokines are known to cause more severe age-related deficiencies in cognitive functions [49]. Recently, it was shown that a combination of CCL-11 with IL-1, IL-1RA, TNF-α, sTNF-R1, sTNFR2 and CCL-2 predicts deficit schizophrenia with a bootstrapped (2000 bootstraps) area under the receiver operating curve of 0.985 [50]. Increased levels of CCL-11 coupled with increased IL-6 and Dickkoph-1-related protein (DKK1) also predict a non-response to treatment with antipsychotics [51].

Most importantly, in schizophrenia, increased levels of CCL-11 strongly impact many neurocognitive tests [9]. Sirivichayakul et al. (2018) established that CCL-11 was highly significantly associated with impairments in many CERAD and CANTAB tests including probes of semantic and episodic memory, as well as executive functions [9]. For example, CCL-11 alone explained 16.0% of the variance in the Verbal Fluency Test (VFT) results and 11.0% of the variance in an index of semantic memory [9]. Interestingly, also formal thought disorders, a key symptom of schizophrenia, were significantly associated with increased levels of CCL-11 [9]. Another study observed highly significant associations between increased CCL-11 levels and cognitive impairments in attention, working memory, episodic and semantic memory and executive functions [50].

Moreover, increased CCL-11 plasma levels are also associated with increased severity scores on different symptom domains of schizophrenia [8,9,50,51,52]. First, in schizophrenia, positive correlations were established between increased CCL-11 levels and negative symptoms [8,9,50,52], but also with psychosis, hostility, excitation, mannerism and psychomotor retardation [9,50,51]. The impact of CCL-11 on these symptoms may be increased by combining CCL-11 levels with other neurotoxic compounds including tryptophan catabolites such as picolinic and xanthurenic acid [11].

Therefore, it was concluded that CCL-11 alone or together with other immune products including TRYCATs, IL-1β, IL-6 and TNF-α, exerts neurotoxic effects on neuronal cells, thereby causing neurocognitive impairments and the symptom domains of schizophrenia [45]. Moreover, such effects may be aggravated by impairments in the Compensatory Immune-Regulatory System (CIRS), including lowered levels of natural IgM directed against oxidative specific epitopes [45].

### 2.5. CCL-11 in Mood Disorders 

After the first study reporting increased levels of CCL-11 in patients with major depression, especially associated with suicidal ideation in 2012, Magalhaes et al. (2014) and Barbosa et al. (2013) also reported increased levels of CCL-11 in patients with bipolar disorder [53,54]. Not only in bipolar disorder, also in persistent depressive disorder or dysthymia, CCL-11 is found to be increased [55]. Simon et al. (2008) assessed serum levels of 22 cytokines/chemokines, including CCL-11, in 49 patients with major depression and 49 matched controls, and reported increased levels of CCL-11 in the context of a “generalized chronic inflammatory state” [56]. Texeira et al. (2018) reported similar results in an independent cohort of patients with major depression, indicating that increased serum levels of CCL-11 were associated with suicidal ideation [52]. Nevertheless, a recent meta-analysis of studies evaluating CCL-11 in depression including 454 participants failed to identify significant differences in CCL-11 between depressed and control subjects [57]. It is possible, however, that this meta-analysis also included patients with milder depression symptoms or different comorbidities explaining the differences between studies [57]. It should be added that a recent paper was unable to detect significant alterations in serum CCL-11 in children with major depression as compared with controls while immune-inflammatory markers were clearly elevated [50]. In cocaine use disorder, CCL-11 combined with TNF-α, IL-1β, CXCL12, CCL-2 and CX3CL1 (C-X3-C motif chemokine Ligand 1) can be used to distinguish primary major depression from substance-induced major depression, indicating that plasma concentrations of CCL-11 may be used as a potential biological marker to differentiate between primary and substance-induced depression [58].

Increased CCL-11 levels may also play a role in Premenstrual Syndrome (PMS), re-labelled as a Menstrual Cycle-Associated Syndrome (MCAS) [45]. Thus, a recent paper shows that CCL-11 is (along with CCL-2 and CCL-5) significantly increased in MCAS as compared with women without MCAS. The increased levels of CCL-11 (and CCL-2, CCL-5, CXCL10 and CXCL8) are highly significantly associated with the severity of MCAS symptoms as measured with the Daily Record of Severity of Problems (DRSP) score [45]. Moreover, the sum of three neurotoxic chemokines (namely CCL-2 + CCL-11 + CCL-5) is significantly associated with the depressive and anxiety subdomains of the MCAS [45]. This is important as MCAS/PMS is one of the predictors of mood disorders including major depression, and therefore, these findings further suggest that CCL-11 may be associated with the pathophysiology of depression (Table 3).

### 2.6. CCL-11 in Other Psychiatric Disorders 

#### 2.6.1. CCL-11 in Obsessive-Compulsive Disorder 

In the study of Fontenelle (2012), forty patients with OCD and 40 healthy controls had their plasma assessed for chemokines including CCL-11 (and CCL-2, CCL-3, CCL-24, CXCL8, CXCL9, CXCL10) and other immune mediators like TNF-α, sTNFR1, sTNFR2 and interleukin-1 receptor antagonist [59]. However, there were no significant differences in the blood levels of CCL-11 between patients with OCD and controls, whereas CCL-3, CXCL8, sTNFR1 and sTNFR2 were significantly increased [59] (Table 3).

#### 2.6.2. CCL-11 in Autism Spectrum Disorder 

There are few studies that show high levels of CCL-11 in ASD [60,61,62]. A meta-analysis reviewed the results of 17 ASD studies with a total sample size of 743 patients with ASD versus 592 healthy controls [62]. This study examined 19 cytokines and reported significantly higher plasma/serum levels of 12 of the cytokines/chemokines in ASD versus controls including CCL-11 (*p* = 0.01), IL-1β (*p* < 0.001), IL-6 (*p* = 0.03), IL-8 (*p* = 0.04), IFN-γ (*p* = 0.02) and CCL-2 (*p* < 0.05), whereas TGF-β1 levels were significantly lower in ASD (*p* < 0.001) [62]. Importantly, increased CCL-11 (and IL-6, IL-10 and MCP-3) were found in the anterior cingulate gyrus and increased CCL-2 and TGF-β1 in the CSF, anterior cingulate gyrus and cerebellum of ASD brain tissues [63] (Table 3).

#### 2.6.3. CCL-11 in Substance Abuse Disorders

Kuo et al. (2018) included 344 heroin-dependent Taiwanese patients under methadone maintenance treatment as compared with 87 normal control subjects in order to investigate plasma CCL-11 and a SNP (Single-Nucleotide Polymorphism) of the CCL-11 gene and Fibroblast Growth Factor-2 (FGF-2). In patients, but not normal controls, CCL-11 showed an adequate sensitivity and specificity in association with age using a cut-off at 45 years whilst increased plasma FGF-2 levels were correlated with the high CCL-11 level [64]. Decreased plasma levels of CCL-11 were observed in 87 abstinent patients with alcohol use disorder as compared with 55 controls [65]. Moreover, subjects with mood disorders and/or anxiety had lower CCL-11 concentrations than non-comorbid patients, and this effect was pronounced in women [65]. Preclinical models of alcohol use in male Wistar rats showed alcohol-induced circulating chemokine alterations in CCL-11, CXCL12 and CX3CL1 [65], indicating an important contribution of CCL-11 to alcohol use disorder (Table 3).

### 2.7. CCL-11 in Neuro-Inflammatory Disorders

#### 2.7.1. CCL-11 and Parkinson’s Disease

Parkinson’s disease is a neuro-inflammatory disorder in which immune-inflammatory processes are involved in the degeneration of dopaminergic neurons via, amongst other mediators, chemokines [66]. Scalzo et al. (2011) found no significant differences in chemokine levels between patients with Parkinson’s disease and controls [66]. Lindqvist et al. (2013) measured immune-inflammatory biomarkers in CSF samples from Parkinson’s disease patients to determine the relationships with non-motor Parkinson’s disease symptoms, and they reported increased levels of CSF CCL-11, C-reactive protein, IL-6 and TNF-α in association with depression, fatigue and cognitive impairments [67]. On the other hand, Moghadam-Ahmadi et al. (2018) concluded that CCL-11 is not involved in the diagnosis or treatment of Parkinson’s disease [68]. In animal models, Chandra et al. (2017) reported that neutralization of CCL-11 and CCL-5 may protect against nigrostriatal neurodegeneration and therefore the progression of Parkinson’s disease [69].

#### 2.7.2. CCL-11 and Alzheimer’s Disease

As mentioned above, CCL-11 is known to cause cognitive decline with age, but less is known regarding its involvement in the pathogenesis of Alzheimer’s diseases. CCL-11 is considered to contribute a probable risk factor for the development of Alzheimer’s disease [37]. CCR3 and CCR5, which are present on microglia, are more expressed by reactive microglia of patients with Alzheimer’s disease than controls [38]. Lalli et al. (2015) identified a haplotype of SNPs on chromosome 17 within a chemokine gene cluster that modifies the age of onset of Alzheimer’s disease and additionally confers a strong protective effect. Importantly, this haplotype disrupts the associations between increasing age and increasing CCL-11 levels, suggesting CCL-11 may be a novel modifier of Alzheimer’s disease [70]. Guerreiro et al. (2015) reported that the association between the age of onset of Alzheimer’s disease and CCL-11 could lead to the development of immunomodulating therapies, which could be used to delay the onset of the disease [71]. Nevertheless, another study showed that CCL-2 is a better biomarker than CCL-11 for the progression of Alzheimer’s disease, although both chemokines employ the same CCR2 signaling pathway for the accumulation of microglia at sites of neuroinflammation [72]. Deleting CCR3 in mice induces a decrease in synaptic loss, as well as a decrease in spatial learning and memory deficits, further suggesting that age-related increments in CCL-11 confer risk to Alzheimer’s disease [37]. Similar findings were reported by Baruch et al. (2013) who showed that increased production of CCL-11 due to destructive Th-2 inflammation is associated with cognitive dysfunctions [16]. Overall, it appears that age-related increases in CCL-11 confer risk for Alzheimer’s disease while antagonizing CCR3 may lead to therapeutic benefits [37].

#### 2.7.3. CCL-11 and Multiple Sclerosis

Multiple Sclerosis (MS) is a chronic autoimmune and neuro-inflammatory disease of the central nervous system characterized by damage to myelinated axons with varying degrees of destruction of myelin and axons [73,74]. In a meta-analysis, which reviewed 226 research papers and 13,526 patients, Bai et al. (2019) found that 13 CSF cytokines (from a list of 26) and 21 blood cytokines (from a list of 37) were elevated in MS as compared with controls and that CSF CCL-11 was significantly increased with a large effect size in patients with MS [75]. Huber et al. (2018) reported that during the relapse phase, CCL-11 may be downregulated, while during the secondary progressive phase, CCL-11 is upregulated to achieve plateau levels [40]. In other studies, it was detected that CCL-11 (and CCL-5) may discriminate clinical subtypes of multiple sclerosis [76,77,78]. Thus, both CCL-11 and CCL-5 were lowered during the inflammatory phase as compared with progressive multiple sclerosis [77]. Michael et al. (2013) identified important chemokine profile markers including CCL-11 in patients with Neuromyelitis Optica (NMO), which were additionally different from those with MS [76]. These authors examined 29 aquaporin antibody-positive NMO patients and found CCL-11 (and CCL-4, G-CSF and myeloperoxidase) to be an important marker in NMO [76]. In NMO Spectrum Disorders (NMOSD), increasing production of TNF-α and interleukin-1β can stimulate CCL-11 binding to CCR3, which may lead to eosinophil hypersensitivity during remission, but not in MS [77]. Moreover, increased CCL-11 levels may be a critical step in NMOSD eosinophil restoration during remission while both elevated levels of CCL-11 and CCL-13 may be important in eosinophil recruitment during NMOSD remission [77,78].

Autoimmune encephalomyelitis (EAE), an experimental model of multiple sclerosis, is accompanied by elevated levels of CCL-11 in the CSF, a tighter blood-brain barrier, reduced antigen-specific responses and a predominant anti-inflammatory Th-2 phenotype, suggesting that CCL-11 may protect against neuroinflammation in this model [79].

#### 2.7.4. CCL-11 and Stroke

CCL-11 is not only an immune marker, which has great importance in aging, neuroinflammation and neurodegeneration, but this chemokine is also associated with stroke [80,81]. Khavinson et al. (2016) reviewed that cardiovascular disorders are accompanied by increased CCL-11 concentrations, and accordingly, these authors proposed to use the term “protein of senility” to describe the detrimental effects of CCL-11, this in contrast to lowered levels of plasma Growth Differentiation Factor (GDF)11, which increases the risk of myocardial infarction and stroke and, therefore, should be considered as a “protein of juvenility” [80]. Bone GDF11 or Morphogenetic Protein 11 (BMP 11) is a Transforming Growth Factor-β (TGF-β) family member protein that is produced by humans, rats and mice [80]. Interestingly, GDF11 has paradoxical effects starting from embryonic development, and the role of GDF11 in aging and medical disease varies throughout the lifespan [81]. A survey performed by Sharma et al. (2014) indicated that CCL-11 may be employed as one of the most important biomarkers for acute stroke even in patients with stroke-like symptoms [81]. Selected from a list of 262 potential biomarkers, the latter authors selected five biomarkers, which were combined in a model built with stepwise selection, validated by bootstrapping and included CCL-11, epidermal growth factor receptor, S100A12, metalloproteinase inhibitor-4 and prolactin. This model could not only be used as an external validating criterion for the diagnosis stroke, but also as a biomarker for the management of stroke and stroke-like symptoms in patients [81]. Roy-O’Reilly et al. (2017) showed lower CCL-11 levels in stroke patients as compared with controls and, additionally, that lower post-stroke CCL-11 levels predict increased stroke severity and poorer functional outcomes 12 months after ischemic stroke [82]. Moreover, a CCL-11 gene polymorphism is associated with different types of ischaemic stroke [83]. A study on various CCL-11 gene variants in Chinese people suggested a strong association with ischaemic stroke, although there were no associations with ischaemic stroke subtypes [84]. Six tag SNPs in the CCL-11 gene (rs1129844, rs17809012, rs1860183, rs1860184, rs4795898 and rs4795895) were studied in 620 patients with stroke and in a control group of 425 Han population patients in China [83]. These authors reported that all polymorphisms of the CCL-11 gene had different effects on the pathogenesis of lacunar stroke [83]. Another large-n study stratified 1500 ischaemic stroke patients into TOAST (Trial of ORG 10172 in Acute Stroke Treatment) subtypes and reported significant associations among the -1382A>G variant of the CCL-11 gene with intracranial large artery atherosclerosis and small vessel occlusion [85].

### 2.8. Possible Novel Treatments Targeting CCL-11

Resveratrol and its metabolites modulate the expression of CCL-11 [86]. In 2011, Yang et al. investigated the effects of resveratrol in modulating inflammation by determining the expression and release of CCL-11 in cultured Human Pulmonary Artery Endothelial Cells (HPAEC) treated with the proinflammatory cytokines IL-13 and TNF-α [86]. Exposure to resveratrol suppressed IL-13- and TNF-α-induced CCL-11 gene expression, as well as attenuated CCL-11 promoter activity, in association with inhibition of Janus Kinase-1 (JAK-1) expression, reduction in phosphorylated-STAT6 and decrements in the p65 subunit of NF-κB [86]. Not only does resveratrol have the ability to modulate CCL-11, but also piceatannol, one of resveratrol’s metabolites, had a similar potency as resveratrol [86]. This in vitro model can be used for further screening and discovering polyphenols with anti-inflammatory activities [86].

A human single-chain fragment variable antibody that neutralizes human CCL-11 (CAT-212) was produced using antibody phage display and converted to whole antibody IgG4 format (CAT-213) [87]. Further optimization entailed a reduction of the length of the variable heavy chain complementarity-determining region 3 by one amino acid resulting in a 1000-fold increase in potency compared with the parent anti-CCL11 antibody [87]. CAT-213 neutralizes the ability of CCL-11 to increase intracellular calcium signaling (with an IC(50) value of 2.86 nM), migration of CCR3-expressing L1.2 cells (with an IC(50) value of 0.48 nM) and inhibition of the CCL-11-evoked shape change of human eosinophils in vitro [87]. CAT-213 and CAT-212 do not bind or neutralize MCP-1, a chemokine with a similar structure [87]. In vivo and in vitro probes conducted by those authors with recombinant human CCL-11 (Cambridge Bioscience and R&D Systems), mouse CCL-11 (R&D Systems) and synthetic human, rat and monkey CCL-11 (Albachem Limited) showed that CAT-212 and CAT- 213 are potent and may be used as a therapy targeting high CCL-11 levels [87]. New clinical trials with anti-CCL11 neutralizing antibodies are expected to have encouraging results in inflammatory diseases [87].

Minocycline treatment can significantly reduce the amounts of several inflammatory factors, including CCL-11 (and CCL-2, MCP-5, IL-6 and IL-10), and therefore, this drug has potent anti-inflammatory and neuroprotective properties in rodent models of Huntington’s, Parkinson’s, Alzheimer’s and motor neuron disease [88].

The glycosaminoglycan heparin has anti-inflammatory activity and is exclusively found in mast cells, which are localized within Airway Smooth Muscle (ASM) bundles of asthmatic airways [89]. IL-13 induces the production of multiple inflammatory mediators from ASM including CCL-11 [89]. Inhibition of IL-13-dependent CCL-11 release by heparin involves, but does not depend on sulphation, though the loss of N-sulphation reduced the attenuating activity, which could be restored by N-acetylation [89].

In vitro studies have shown an inhibitory effect of glucocorticosteroids on CCL-11 production, although there are few in vivo studies [90,91,92]. The use of glucocorticosteroids to inhibit CCL-11 mRNA expression was studied by Jahnsen et al. (1999) who examined transcript levels and chemotactic activity of CCR3-binding chemokines in nasal polyps. Treatments with glucocorticosteroids reduce mRNA levels in polyps to levels that are found in turbinate mucosa for all chemokines [90]. Whether this result was caused indirectly via extracellular mediators that regulate chemokine production or by a direct effect of glucocorticoids could not be determined. Nevertheless, glucocorticoids may directly inhibit cytokine-induced production of CCL-11 (and MCP-4 and CCL5) in epithelial cells in vitro [91,92].

Interestingly, the detrimental effects of CCL-11 suppressing hippocampal neurogenesis can be attenuated by supra-lactate threshold exercise, hence supporting the benefits of exercise on the aging brain [93].

## 3. Materials and Methods

We searched online libraries, including PubMed/MEDLINE, Google Scholar and Scopus. The main search terms were “Chemokine CCL11” [MeSH] (Medical Subject Headings) or “biomarkers” [MeSH] and “stress” [MeSH]” and “Chemokine CCL11” [MeSH], “schizophrenia” and “CCL11” [MeSH] and “brain” [MeSH] with filters activated, namely publication date from 01/01/1990 to 31/12/2019 and papers written in English. Thus, our references were 94 (ninety-four).

## 4. Conclusions 

Increased CCL-11 production in the course of immune-mediated inflammatory conditions may play a role in age-related cognitive decline, schizophrenia, mood disorders, Alzheimer’s disease, OCD and ASD. Plasma levels of CCL-11 are increased not only in schizophrenia and age-related cognitive impairments, but also in some patients with mood disorders and premenstrual syndrome. Increased CCL-11 levels including in old age are associated with neurodegeneration, reduced neurogenesis and an increased risk of Alzheimer’s disease. Increased CCL-11 also plays a role in neuroinflammatory disease including multiple sclerosis. Therefore, increased CCL-11 is a new drug target for the treatment and prevention of those disorders. The production of CCL-11 may be attenuated by glucocorticoids, minocycline, resveratrol, CCR3 antagonists (R321) and anti-CCL11 antibodies including CAT-212 and CAT-213. In sum, these findings show real encouraging results for future treatment of all conditions characterized by increased CCL-11 levels (Figure 1).

## Figures and Tables

**Figure 1 pharmaceuticals-13-00230-f001:**
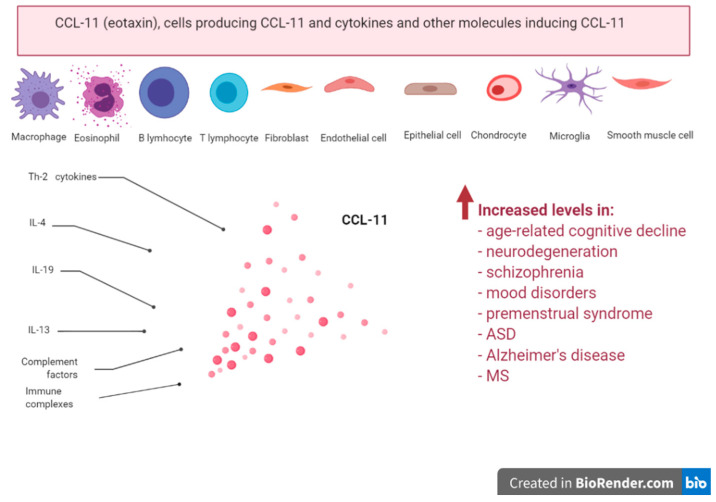
Take-home message; producers and inducers of CCL-11; increased levels in age-related cognitive decline, neurodegeneration, schizophrenia, mood disorders, premenstrual syndrome, ASD, Alzheimer’s disease and multiple sclerosis (MS).

**Table 1 pharmaceuticals-13-00230-t001:** Cells producing CCL-11.

Cells Producing CCL-11 [8,9,10]
Eosinophils
Macrophages
T and B-cells
Fibroblasts
Endothelial cells
Epithelial cells
Chondrocytes
Microglia
Keratinocytes
Smooth muscle cells

**Table 2 pharmaceuticals-13-00230-t002:** Cytokines and other molecules inducing CCL-11.

Cytokines and Other Molecules Inducing CCL-11	References
Th-2 cytokines	[8] (Teixeira AL et al., 2018) [9] (Sirivichayakul S et al., 2018) [10] (Kindstedt E et al., 2017)
Interleukin IL-4	
Interleukin IL-10	
Interleukin IL-13	
Complement factors	
Immune complexes	

**Table 3 pharmaceuticals-13-00230-t003:** Studies on CCL-11 regarding different psychiatric disorders.

Psychiatric Disorder	Findings for CCL-11	References
Schizophrenia	Increased blood levels; negative correlation with telomere length and grey matter volume; negative correlation with cognitive measures; positive correlation with negative symptoms.	[52] (Teixeira et al., 2008) [47] (Czepielewski et al., 2018) [46] (Hong et al., 2017) [50] (Al-Hakeim et al., 2019) [51] (Al-Dujaili et al., 2019)
Deficit schizophrenia	Increased plasma CCL-11 levels	[9,11] (Sirivichayakul et al., 2018; 2019)
Bipolar disorder	Increased blood levels; association with illness stage.	[54] (Barbosa et al., 2013)
Major depression	Increased blood levels; association with suicidal ideation.	[56] (Simon et al., 2008)
Dysthymia	Increased blood levels.	[55] (Ho et al., 2017) [53] (Magalhaes et al., 2014)
Premenstrual syndrome	Increased plasma CCL-11.	[45] (Roomruangwong et al., 2019)
OCD (Obsessive-Compulsive Disorder)	Blood levels similar to controls.	[59] (Fontenelle et al., 2012)
Autism spectrum disorder	Increased blood levels; increased CCL-11, IL-6, IL-10 and MCP-3 in the anterior cingulate gyrus in ASD brain specimens.	[60] (Ashwood et al., 2006) [61] (Cunha et al., 2015) [62] (Masi et al., 2015) [63] (Zimmerman et al., 2005)
Substance abuse disorder	In heroin-dependent subjects, increased blood levels and association with age; in alcohol-dependent subjects, decreased blood levels, especially in women and with comorbid psychiatric disorders.	[64] (Kuo et al., 2018) [65] (Garcia-Marchena et al., 2016)

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
