# Peer review of "CCL-11 or Eotaxin-1: An Immune Marker for Ageing and Accelerated Ageing in Neuro-Psychiatric Disorders"

_pharmaceuticals, 2020, doi:10.3390/ph13090230_

Round 1

Reviewer 1 Report

The authors have submitted a manuscript of illustrating a current overview regarding possible therapeutic regulation of increased levels of CCL11 (i.e. eotaxin, one of the chemokines identified) on neurodegenerative diseases such as Alzheimer’s disease, Parkinson’s disease, schizophrenia, mood disorders, and other psychiatric disorders. The authors searched a range of eligible literature, from well-known neural and molecular basis of pathology for neurodegenerative diseases to current knowledge of CCL11 pathway for a possible target of neuroprotection, resulting in reliable conclusions. The authors described reported pharmacological properties of some chemical substances such as antibiotic tetracyclines and glucocorticoids on CCL11 production which may affect neurodegenerative diseases. This issue is of interest, and impact of their review is strong. My overall concern with the review describing the current available data regarding pharmacotherapeutic option of CCL11 pathway is that information provided may offer something substantial that helps advance our understanding of novel medicines against neurodegenerative diseases. The reference list may be useful for readers who are interested in this issue. However, the authors should consider the following issues in their revised manuscript to be accepted for publication.

Major issues:

To better understand the possibility of the CCL11 pathway usefulness, readers need to easily know the take-home messages from this review article: why and how the CCL11 pathway could be considered to cure patients with neurodegenerative diseases. For instance, a schematic diagram of putative molecular mechanisms of neurodegenerative diseases would strengthen this manuscript. The opposite, adverse effects of CCL11 pathway except neurodegenerative diseases, if known, may influence largely the authors’ perspective. The authors are strongly recommended to add the take-home message, especially a novel figure illustrating the actions of the CCL11 pathway in this review.

Minor points:

  1. The authors have used several abbreviations for eotaxin, lacking in unity: CCL11, CCl11, and CCL-11. Please use an uniform abbreviation for eotaxin throughout the manuscript.

  1. Methods section: Please describe clearly how many references the authors have collected according to their searches within the databases, and their criteria which determined the selection of the reference which were included in this manuscript. These are important information for readers to better understand and evaluate the review articles.

Author Response

Dear Reviewers,Thank you for giving us the opportunity to submit a revised draft of the manuscript “CCL-11 or Eotaxin-1: An Immune Marker for Ageing and Accelerated Ageing in Neuro-Psychiatric Disorders” . Weappreciate the time and effort that you and the reviewers dedicated to providing feedback on ourmanuscript and are grateful for the insightful comments on and valuable improvements to our paper.We have incorporated most of the suggestions made by the reviewers. Please see below, in red, for a point-by-point response to the reviewers’comments and concerns.

Reviewer 1:

Major issues: To better understand the possibility of the CCL11 pathway usefulness, readers need to easily know the take-home messages from this review article: why and how the CCL11 pathway could be considered to cure patients with neurodegenerative diseases. For instance, a schematic diagram of putative molecular mechanisms of neurodegenerative diseases would strengthen this manuscript. The opposite, adverse effects of CCL11 pathway except neurodegenerative diseases, if known, may influence largely the authors’ perspective. The authors are stronglyrecommended to add the take-home message, especially a novel figure illustrating the actions of the CCL11 pathway in this review.

Response 1: Thank you for this suggestion. As suggested by the reviewer, we havemade the change and we upload a figure 1 showing the producers, inducers of CCL-11and in which cases we have increase levels of CCL-11. The figure can be seen in the revised manuscript lines 443-447.

Minor points:1.The authors have used several abbreviations for eotaxin, lacking in unity: CCL11, CCl11, and CCL-11. Please use an uniform abbreviation for eotaxin throughout the manuscript.

Response 2:Thank you for pointing this out. Thechange has been made in the whole document. We use everywhere CCL-11.

2.Methods section: Please describe clearly how many references the authors have collected according to their searches within the databases, and their criteria which determined the selection of the reference which were included in this manuscript. These are important information for readers to better understand and evaluate the review articles.

Response 3: We think this is an excellent suggestion. We have made the change. We include the exact number of references included in the study (lines 426-431).We searched online libraries, including PubMed/MEDLINE, Google Scholar, and Scopus. The main search terms were “Chemokine CCL11” [MeSH] or “biomarkers” [MeSH] and “stress” [MeSH]” and “Chemokine CCL11” [MeSH], “schizophrenia”, and “CCL11” [MeSH] and “brain” [MeSH] with filters activated, namely publication date from 01/01/1990 to 31/12/2019 and papers written in English. Thus, our references are 94 (ninety-four).

Reviewer 2 Report

The authors reviewed the associations between CCL11 (eotaxin) and aging, neurodegenerative, neuroinflammatory and neuropsychiatric disorders.

This review is interesting and new, as well as contains the important information about the CCL11 as a new drug target in the treatment and prevention of many human disease including age-related cognitive decline, schizophrenia, mood disorders and neurodegenerative disorders.

Nevertheless, this manuscript needs corrections and improvements before publishing is possible.

General points:

Please add list of abbreviations to your manuscript.

Please check and correct all titles and also subtitles in the whole manuscript: sometimes you did use a capital letters and sometimes the small letters.

Please add additional Figures to your manuscript: for example, with the molecular structure of CCL11 or with the regulatory mechanisms of CCL11.

Special points:

Keywords: please add also to keywords: eotaxin; Alzheimer’s disease; aging; Stroke; Schizophrenia; biomarker; prevention  

Introduction

Lines 50-51: please add references at the end of this sentence.

Line 55: please say:  Central nervous system (CNS).

Lines 56-58:  please add references at the end of this sentence.

Lines 58-60: please add references at the end of this sentence.

Table 1: Please show as separately the “Cells producing CCL11” as a Table 1. and separately “Cytokines and other molecules inducing CCL11 as a separately Table 2.    Please add as a last column (right side) a references numbers according to your List of references like as Table 2.

Lines 72-77: please describe as exactly which neurodegenerative disease are meant.  

Results and Discussion

Lines 83-84: please add references at the end of this sentence.

Lines 105-107: please add references at the end of this sentence.

Lines 158-160: please add references at the end of this sentence.

Lines 165-167: please add references at the end of each these sentences.

Lines 172-173: please add references at the end of this sentence.

Lines 187-188: please add references at the end of this sentence.

Lines 188-190: please add references at the end of this sentence.

Lines 196-197: please add references at the end of this sentence.

Lines 202-204: please add references at the end of this sentence.

Lines 218-220: please add references at the end of this sentence.

Lines 230-232: please add references at the end of this sentence.

Lines 270-272: please add references at the end of this sentence.

Lines 310-311: please add references at the end of this sentence.

Lines 327-328: please add references at the end of this sentence.

Lines 333-335: please add references at the end of this sentence.

Lines 335-337: please add references at the end of this sentence.

Line 357: please add references at the end of this sentence.

Lines 364-366: please add references at the end of this sentence.

Lines 379-380: please add references at the end of this sentence.

Lines 385-386: please add references at the end of this sentence.

Lines 391-392: please add references at the end of this sentence.

Supplementary Materials: what about the Tables 1 and 2? Are they a part of the manuscript or they are a part of the Supplement? Please keep both Tables in the body of the manuscript.

Please correct your List of references at the end of the manuscript according to Pharmaceuticals.

Please see an example, one publication recently published in “Pharmaceuticals”:

Review

A Systematic Review of Molecular Imaging Agents

Targeting Bradykinin B1 and B2 Receptors

Joseph Lau 1, Julie Rousseau 1 , Daniel Kwon 1, François Bénard 1,2,* and Kuo-Shyan Lin 1,2,*

1 Department of Molecular Oncology, BC Cancer, Vancouver, BC V5Z 1L3 Canada; lauj2@nih.gov (J.L.);

jrousseau@bccrc.ca (J.R.); dkwon@bccrc.ca (D.K.)

2 Department of Radiology, University of British Columbia, Vancouver, BC V6T 1Z4, Canada

* Correspondence: Canada fbenard@bccrc.ca (F.B.); klin@bccrc.ca (K.-S.L.); Tel.: +1-604-675-8206 (F.B.);

+1-604-675-8208 (K.-S.L.); Fax: +1-604-675-8218 (F.B. & K.-S.L.)

Pharmaceuticals 2020, 13, 199; doi:10.3390/ph13080199

By the way, you can see also an example for Figures and Tables.  

Author Response

Dear Reviewers,Thank you for giving us the opportunity to submit a revised draft of the manuscript “CCL-11 or Eotaxin-1: An Immune Marker for Ageing and Accelerated Ageing in Neuro-Psychiatric Disorders” . We appreciate the time and effort that you and the reviewers dedicated to providing feedback on ourmanuscript and are grateful for the insightful comments on and valuable improvements to our paper.We have incorporated most of the suggestions made by the reviewers. Please see below, in red, for a point-by-point response to the reviewers’comments and concerns.

Reviewers' Comments to the Authors:

Reviewer 2:The authors reviewed the associations between CCL11 (eotaxin) and aging, neurodegenerative, neuroinflammatory and neuropsychiatric disorders.This review is interesting and new, as well as contains the important information about the CCL11 as a new drug target in the treatment and prevention of many human disease including age-related cognitive decline, schizophrenia, mood disorders and neurodegenerative disorders.Nevertheless, this manuscript needs corrections and improvements before publishing is possible.

Response1:Thank you!

General points:Please add list of abbreviations to your manuscript.

Response2: We think this is an excellent suggestion. We havemade a list of abbreviations. Those are lines 455-456 in the revised manuscript.

Please check and correct all titles and also subtitles in the whole manuscript: sometimes you did use a capital letters and sometimes the small letters.

Response 3: Thank you for pointing this out. Thecapital and small letters are corrected. You can use track changes on our document to see all changes made.

Please add additional Figures to your manuscript: for example, with the molecular structure of CCL11 or with the regulatory mechanisms of CCL11.

Response 4: Thank you for this suggestion. As suggested by the reviewer, we havemade the change and we upload a figure 1 showing the producers, inducers of CCL-11and in which cases we have increasedlevels of CCL-11. The figure can be seen in the revised manuscript lines 443-447.

Special points:Keywords: please add also to keywords: eotaxin; Alzheimer’s disease; aging; Stroke; Schizophrenia; biomarker; prevention

Response 5:We have added the suggested content to the manuscript onlines 53-54 in the revised manuscript.

Keywords: brain, behavior, cytokines, CCL-11, eotaxin, Alzheimer’s disease, aging, stroke, schizophrenia, biomarkers, prevention

Introduction

Lines 50-51: please add references at the end of this sentence.

Line 55: please say: Central nervous system (CNS).

Lines 56-58: please add references at the end of this sentence.

Lines 58-60: please add references at the end of this sentence

Response6:Lines 50-51, 55, 56-58 and 58-60 have been updated, such that references are added to the end of the sentenceand we wrote central nervous system.

Chemokines consist of a family of small cytokines (712 kDa), which prompt directed chemotaxis in nearby responsive cells. Chemokines play a pivotal role in immune functions and inflammatory responses and facilitate leukocyte migration and trafficking [1-3]. The discovery of chemokines goes back to 1977 [4-5], but their role in altering the neuroimmune and neurobiological processes gained notice until the mid-90s [6]. Recent studies show a direct role of chemokines in neuroendocrine function, neurotransmission, and neurodegeneration within the CNS (Central nervous system) [7]. Chemokines comprise four families characterized according to the relative position of their cysteine residues and their functions, with CCL (C-C motif chemokine) and CXCL (C-X-C motif ligand) being the largest [7]. They act by binding to seven-transmembrane G protein-coupled receptors, which in turn activate signaling cascades and initiate shape rearrangement and cell movement [7].

Table 1: Please show as separately the “Cells producing CCL11” as a Table 1. and separately “Cytokines and other molecules inducing CCL11 as a separately Table 2. Please add as a last column (right side) a references numbers according to your List of references like as Table 2.

Response 7:Thank you for pointing this out. The change has been made. We made two separate tables. Table 1. Cells producing CCL-11(lines 75-76 in the revised manuscript).

Also we made Table 2. Cytokines and other molecules inducing CCL-11(lines 77-78 in the revised manuscript)

Lines 72-77: please describe as exactly which neurodegenerative disease are meant.

Response 7: As suggested by the reviewer, we havemade this change. We added an example of neurodegenerative disease (lines 81-86 in the revised manuscript).Increased levels of CCL-11 have been detected in numerous neuro-inflammatory disorders such as multiple sclerosis [13], as well as neurodegenerative and neuroprogressive disorders including Alzheimer’s disease [8] and psychiatric illnesses including major depression, bipolar disorder and schizophrenia [8, 11, 13-15]. Moreover, increased CCL-11 levels are also associated with neurocognitive deficits in aging, neurodegenerative disorders, and major psychiatric disorders such as schizophrenia [11]

Results and Discussion

Lines 83-84: please add references at the end of this sentence.

Lines 105-107: please add references at the end of this sentence.

Lines 158-160: please add references at the end of this sentence.

Lines 165-167: please add references at the end of each these sentences.

Lines 172-173: please add references at the end of this sentence.

Lines 187-188: please add references at the end of this sentence.Lines 188-190: please add references at the end of this sentence.

Lines 196-197: please add references at the end of this sentence.

Lines 202-204: please add references at the end of this sentence.

Lines 218-220: please add references at the end of this sentence.

Lines 230-232: please add references at the end of this sentence.

Lines 270-272: please add references at the end of this sentence

Lines 310-311: please add references at the end of this sentence.

Lines 327-328: please add references at the end of this sentence.

Lines 333-335: please add references at the end of this sentence.

Lines 335-337: please add references at the end of this sentence.

Line 357: please add references at the end of this sentence.

Lines 364-366: please add references at the end of this sentence.

Lines 379-380: please add references at the end of this sentence.

Lines 385-386: please add references at the end of this sentence.

Lines 391-392: please add references at the end of this sentence

Response 8: We thinkthis is an excellent suggestion. We have made theneeded changes. In order not to write all changed lines here, it better to use track changes in our revised manuscript.

Example for changed lines 385-386; 391-392:The glycosaminoglycan heparin has anti-inflammatory activity and is exclusively found in mast cells, which are localized within airway smooth muscle (ASM) bundles of asthmatic airways [90]. IL-13 induces the production of multiple inflammatory mediators from ASM including CCL-11 [90]. Inhibition of IL-13-dependent CCL-11 release by heparin involves but does not depend on sulphation, though the loss of N-sulphation reduced the attenuating activity, which could be restored by N-acetylation [90].In vitro studies have shown an inhibitory effect of glucocorticosteroids on CCL-11 production although there are few in vivo studies [91, 92, 93}. The use of glucocorticosteroids to inhibit CCL-11 mRNA expression was studied by Jahnsen et al. (1999) who examined transcript levels and chemotactic activity of CCR3-binding chemokines in nasal polyps. Treatments with glucocorticosteroids reduce mRNA levels in polyps to levels that are found in turbinate mucosa for all chemokines [91]

Supplementary Materials: what about the Tables 1 and 2? Are they a part of the manuscript or they are a part of the Supplement? Please keep both Tables in the body of the manuscript.

Response 9: Thank you for pointing this out. The reviewer is correct, and we have erased the section supplementary materials. The tables and the figure are part of the manuscript. The revised textis with 3 tables and one figure inserted in the manuscript. Thank you once again.

Please correct your List of references at the end of the manuscript according to Pharmaceuticals.Please see an example, one publication recently published in “Pharmaceuticals”

Review

A Systematic Review of Molecular Imaging AgentsTargeting Bradykinin B1 and B2 ReceptorsJoseph Lau 1, Julie Rousseau 1 , Daniel Kwon 1, François Bénard 1,2,* and Kuo-Shyan Lin 1,2,*1 Department of Molecular Oncology, BC Cancer, Vancouver, BC V5Z 1L3 Canada; lauj2@nih.gov (J.L.);jrousseau@bccrc.ca (J.R.); dkwon@bccrc.ca (D.K.)2 Department of Radiology, University of British Columbia, Vancouver, BC V6T 1Z4, Canada* Correspondence: Canada fbenard@bccrc.ca (F.B.); klin@bccrc.ca (K.-S.L.); Tel.: +1-604-675-8206 (F.B.);+1-604-675-8208 (K.-S.L.); Fax: +1-604-675-8218 (F.B. & K.-S.L.)Pharmaceuticals 2020, 13, 199; doi:10.3390/ph13080199

By the way, you can see also an example for Figures and Tables.

Response 9: We agree with the reviewer’s assessment. Accordingly, throughout the manuscript,we have revised our references according to Pharmaceuticals.

Also, we used as an example some ofthe tables and figures in published manuscripts.References, lines 457-760 in the revised manuscript.

Also, we updated this part:

ReviewCCL-11 or Eotaxin-1: An Immune Marker for Ageing and Accelerated Ageing in Neuro-Psychiatric Disorders

Mariya Ivanovska 1,2,3*, Zakee Abdi 4, Marianna Murdjeva 1,2,3, Danielle Macedo 5,6, Annabel Maes 7, Michael Maes 8,9,10

1 Department of Microbiology and Immunology, Faculty of Pharmacy; Research Institute, Medical University-Plovdiv, 4002, Bulgaria; mariya_ivanovska@mu-plovdiv.bg(M.I.); mariana_murdzheva@mu-plovdiv.bg(Ma.Mu.);

2Laboratory of Clinical Immunology, University Hospital “St. George”-Plovdiv, 4002, Bulgaria; mariya_ivanovska@mu-plovdiv.bg(M.I.); mariana_murdzheva@mu-plovdiv.bg(Ma.Mu);

3Division of Immunological assessment of post-traumatic stress disorder, TCEMED, Plovdiv, 4002, Bulgaria; mariya_ivanovska@mu-plovdiv.bg(M.I.); mariana_murdzheva@mu-plovdiv.bg(Ma.Mu.);44th year medical student, Medical Faculty, Medical University-Plovdiv, 4002, Bulgaria; zabdi08@googlemail.com(Z.A.); 5Neuropsychopharmacology Laboratory, Drug Research, and Development Center, Faculty of Medicine, Universidade Federal do Ceará, Fortaleza, CE, Brazil; daniellesilmacedo@gmail.com(D.M.); 6National Institute for Translational Medicine (INCT-TM, CNPq), Ribeirão Preto, Brazil; daniellesilmacedo@gmail.com(D.M.); 7Janssen PharmaceuticaNV, Beerse, Belgium; AMaes19@its.jnj.com(A.M.); 8Department of Psychiatry, Chulalongkorn University, Bangkok, Thailand; dr.michaelmaes@hotmail.com(Mi.Ma.); 9TCEMED, Medical University-Plovdiv, Plovdiv, 4002, Bulgaria; dr.michaelmaes@hotmail.com(Mi.Ma.); 10 IMPACT Strategic Research Center, Deakin University, Geelong, Australia; dr.michaelmaes@hotmail.com(Mi.Ma.);*Correspondence: Bulgaria, marijaku87@yahoo.com; mariya.ivanovska@mu-plovdiv.bg(M.I.) Tel: +359897312847 Received: date; Accepted: date; Published:

Round 2

Reviewer 1 Report

The authors have addressed properly all the issues raised by reviewers including me. I have no more comments, and now recommend that this manuscript is acceptable for publication in Pharmaceuticals.

Reviewer 2 Report

This manuscript was corrected according to my previously suggestions.